# Initial Impact of Perinatal Loss on Mothers and Their Partners

**DOI:** 10.3390/ijerph20021304

**Published:** 2023-01-11

**Authors:** Laia Delgado, Jesus Cobo, Cristina Giménez, Genís Felip Fucho-Rius, Stephanie Sammut, Laia Martí, Cristina Lesmes, Salut Puig, Noemí Obregón, Yolanda Canet, Diego J. Palao

**Affiliations:** 1Mental Health Department, Corporació Sanitària Parc Taulí, Universitat Autònoma de Barcelona, CIBERSAM, 08208 Sabadell, Spain; 2Department of Psychiatry and Forensic Medicine, Universitat Autònoma de Barcelona, 08193 Bellaterra, Spain; 3Institut d’Investigació i Innovació Parc Taulí (I3PT), CERCA, 08208 Sabadell, Spain; 4Perinatal Mental Health Program, Cerdanyola-Ripollet Outpatient Department, Sant Joan de Dèu Serveis de Salut Mental, 08291 Ripollet, Spain; 5Gynaecology and Obstetrics Department, Hospital Universitari Parc Taulí, Corporació Sanitària Parc Taulí, Universitat Autònoma de Barcelona, 08208 Sabadell, Spain; 6Department of Medicine, Universitat Autònoma de Barcelona, 08193 Bellaterra, Spain

**Keywords:** perinatal losses, mothers, fathers, partners, grief, gender, outcomes

## Abstract

(1) Background: Perinatal Loss affects one in ten women worldwide. It is known to have a deep impact on the physical and psychological wellbeing of the mother. Moreover, there is a lack of information in regard to gender differences. The role of culture, environment, personal characteristics, and gender is yet to be determined in most reports; (2) Objective and Methods: Our aim is to study the initial impact of perinatal losses in an unselected sample of couples, focusing on gender differences. We conducted a longitudinal prospective study with 29 mothers and 17 fathers. Upon discharge from the hospital, they filled out the Edinburgh Postnatal Depression Scale (EPDS), among others. After one-month post-loss, they performed the EPDS and the Short Version of the Perinatal Grief Scale. We used descriptive statistics for the sample and non-parametric tests for the comparison of gender; (3) Results: We found no gender differences in initial depressive symptoms, nor in depressive symptoms, perinatal grief symptoms, or grief level (total scores or complicated grief) one month after the loss; (4) Conclusions: we need to better understand the psychological evolution of couples in cases of perinatal loss without falling into preconceived ideas about the influence of gender.

## 1. Introduction

The World Health Organization (2006) has established that perinatal death includes the period from 22 weeks’ gestation to the first 28 days after birth. Some authors, however, extend the definition of perinatal loss to that occurring at any time during gestation up to the first month of life, as well as to abortions, selective reduction, the death of a twin, the death of a premature infant and bereavement when a newborn is placed for adoption, among others [1]. In Catalonia, perinatal mortality is 4.74 per 1000 newborns [2]. Despite its relative frequency, it remains a little-known issue; society is reluctant to address it, and this can result in families not recognizing it and even tending to hide it [3].

Fetal loss is a very common public health problem and affects one in ten women in their lifetime [4]. Depending on the source, first-trimester losses may occur in 10–25% of all diagnosed pregnancies in developed countries. Approximately 50% of early losses are due to genetic (chromosomal) defects, but may also be due to other factors or unknown causes [5,6]. In fact, previous studies have detected different risk perspectives depending on the population studied, the country of origin or the characteristics of the samples [6,7,8,9,10].

Moreover, fetal loss is a serious, complex health problem. As referred to by Professor Siobhan Quenby and collaborators [11], the consequences of fetal loss can be both physical, such as hemorrhage or infection, and psychological. The psychological consequences include a significant presence of anxiety and depression symptomatology, and can even lead to post-traumatic stress disorder and other severe mental disorders. It can affect some women more than once in their lives, with significant consequences later in life. These experiences are always traumatic, and must be recognized in order to achieve a substantial recovery and prognosis. Moreover, many of these consequences can be experienced with shame, in silence—or worse, silenced. In addition, perinatal death affects family dynamics and the social environment of the parents is severely altered after perinatal death [12]. There is still a stigma, a concealment of the health problems surrounding the brave, “always happy” conception of pregnancy and upbringing. No one wants to hear bad news.

With respect to other grief processes, grief due to gestational loss has its own characteristics. For example, women who have suffered miscarriages do not have physical memories of the lost child—there is no “object to grieve”—and other people are unaware of the woman’s loss. Moreover, the grief process after miscarriage is more rapid than after other losses [13,14,15,16,17]. In every case, the loss of a child is an understandably shocking life event; it involves significant losses in many aspects such as loss of identity, roles, expectations and future projects, among others [18]. Personal characteristics and the specific origin of the losses could influence their emotional and psychopathological impact or the evolution of the grief [3,17,19,20,21,22].

Perinatal bereavement is characterized by its low visibility and recognition. In the past decades, more protocols have been developed to address this vital time. Despite the strategies implemented to improve care for these families, most health centers do not have clear protocols for assistance [23]. Gestational-loss care has been recognized as one of the most neglected areas in health systems worldwide [24,25]. Even so, there are still relatively few studies that explain the evolution of this type of bereavement—especially in terms of possible gender differences.

The majority of guidelines focus primarily on the experiences of heterosexual women, with few to no recommendations pertaining specifically to the experiences of bereaved men. We used to think that women grieved more deeply than men [26], but recent research suggests that men’s grief may differ in style rather than intensity to that of women [27]. Individual, interpersonal, community and public/political factors have been identified as moderators in such differences. Additionally, the normative gender expectations (role of supporter and protector to their female partner) in heterosexual relationships can lead to more hidden or private forms of grief [27], which can lead to even more of a lack of recognition at all levels [28].

The objective of our analysis was to describe the initial impact of perinatal loss in adult people, with a focus on gender differences in a prospective sample attending a specific Perinatal Loss Clinical Program. Our hypothesis was that the initial impact could be quite similar in mothers and fathers.

## 2. Materials and Methods

### 2.1. Design and Setting

Prospective repeated-measures design study of women and men suffering perinatal losses who were assisted in a university hospital at the Valles Occidental Area of Barcelona (Spain).

### 2.2. Participants, Inclusion and Exclusion Criteria

We included male and female participants over 18 years old who were assisted at the Perinatal Loss Clinical Program of our hospital. The inclusion criteria included only the loss of their pregnancy in our own hospital and a signed acceptance to participate in the prospective study. The exclusion criteria included language barriers, mental disabilities and refusing to participate in the study, as well as parents in whom the high-level bereavement due to their loss was considered too intense to proceed with the study. We decided to exclude couples with high bereavement levels due to their loss for ethical reasons. In our experience, emotional support for severe bereavement couples can be disturbed by research proposals, inquiries not directly related to the process or unnecessary administrative demands. All these aspects could affect the emotional perception of the staff support by the couples. There are also some concerns about the capacity to sign the informed consent form in situations of severe emotional shock. There were no exclusions related to the week of the loss or other clinical characteristics. The participants’ characteristics are shown in Table 1.

### 2.3. Ethics

The study was conducted according to the tenets established by the Declaration of Helsinki and Good Research Practice Guidelines. The Health Care Ethics Committee and the Clinical Research Ethics Committee of the Parc Taulí Hospital approved the study protocol. Participants were given written information and were informed about the motivations and implications of the study prior to consenting to participate. Confidentiality during completion of the questionnaire was guaranteed. During the analysis, the identifying data of the participants was anonymized and unlinked.

### 2.4. Procedures

After clinically assessing the emotional state of the parents based on our clinical experience and psychopathological evaluation (we did not use self-administered measures at this point), participants who complied with the inclusion and did not meet the exclusion criteria were invited to participate by the members of the research team. The researchers explained the aims and implications of the study. The participants were informed about the objectives of the study and, after signing the informed consent, they responded to the questionnaires. Notwithstanding the instruments being self-reported, the researchers supported the participants during completion of the form in order to respond to doubts and to give support and relief in case of emotional reactions. The recruitment of cases took place between 26 April 2018 and 28 November 2021 through consecutive sampling.

### 2.5. Sociodemographic Data

Sociodemographic information was collected in a specific ad hoc questionnaire. It included, among others, age, place of origin, level of education, employment, religious beliefs, number of previous pregnancies, the week of gestation at the time of fetal loss and previous experiences of fetal loss.

### 2.6. Instruments

After collecting the socio-demographic variables, on discharge from the hospital (visit 0) we conducted different psychopathological scales, including the Spanish version of the Edinburgh Postnatal Depression Scale (EPDS) [29]. After 1 month (± week) after the fetal loss (visit 1), we conducted the Edinburgh Postnatal Depression Scale (EPDS) and the Spanish for Spain version of the Perinatal Grief Scale (PGS-SV-SP) [30,31,32].

The Spanish version of the Edinburgh Postnatal Depression Scale (EPDS) [29] is a self-administrated scale. It includes 10 items. A score cut-off point of 10/11 fulfilled the diagnosis criteria for major depression. A cut-off of 8/9 identified all the women with major and minor postpartum depression, but had a low positive predictive value (48%). Increasing the cut off to 10/11 identified 100% of women with major depression, resulting in a combined sensitivity of 79%. The maximum score is 30. The duration of the application is about 5 min.

The Perinatal Grief Scale—Spanish version for Spain (PGS-SV-SP) [30,31,32] is a self-administered scale for the assessment of the intensity and impact of grief secondary to a perinatal loss. It includes 16 items (shorter than the original PGS), scoring between 1 (strongly disagree) and 5 (strongly agree). The target population is Spanish women who have suffered a perinatal loss in the last 5 years. The higher the score, the greater the intensity of the grief. Active grief is considered to be present when the sum of the score obtained in items 1 to 4 is equal to or greater than 8 points. Complicated grief is considered possible when the sum of the scores obtained in items 5 to 16 is equal to or greater than 42 points. In addition, the participant will be considered to be experiencing high-intensity grief when the sum of the total score on the scale is equal to or greater than 49 points.

### 2.7. Statistical Analysis

Descriptive statistics are presented as medians and standard deviation (SD) or as proportions (percentage). For comparison of ordinal quantitative variables from two independent samples we used the Mann–Whitney U test. For comparisons of qualitative variables we used the Chi Square test. Analyses were conducted using the statistics package SPSS v.21.0 for Windows.

## 3. Results

### 3.1. Participants

A total of 41 couples (all of them composed of a mother and a father) were considered candidates for the study. Several cases did not fulfill the inclusion criteria: a couple refused to participate in the study due to religious motivations, four couples accepted verbally to participate in the study, but they did not sign the informed consent in the end. Another couple was heavily bereaved at the time of their loss (they did not want to communicate with the medical staff at this time) and, in the end, we decided not to include them at the sample. A mother suffering from mental disability was also retired from the study. Regarding partners, two fathers signed the informed consent, but they did not complete the evaluation. In eleven cases it was not possible to locate the father. Other participants did not fill out all the scales or give all the required information.

In the end, the initial sample included 29 women and 17 men (17 couples). One month after the loss, we obtained completed information from 18 participants—11 mothers and 7 fathers (seven couples).

### 3.2. Sociodemographic Characteristics

Table 1 indicates the participants’ sociodemographic characteristics, including their age, pregnancy weeks, number of previous children, country of origin, educational level, marital status, work situation, religious belief and previous experiences of perinatal loss. Most of our participants were from Catalonia and other areas of Spain, but some of them were also from Central and South America and other countries of the world. Their educational levels were mainly secondary, and most of the participants were in a qualified job. Most of the participants identified themselves as Christians and 31% of the mothers had previous experiences of perinatal losses.

There were no gender differences related to age, number of previous children, marital state, country of origin, educational level, work situation or religious beliefs (Table 1).

#### Sociodemographic Characteristics by Gender at Visit 0

There were no significant gender differences in sociodemographic characteristics (Table 1).

### 3.3. Psychopathological Impact by Gender

#### 3.3.1. Psychopathological Impact according to Gender at Visit 0

Table 2 indicated the psychopathological impact according to gender. At 24–48 h after the loss, median scores in the Edinburgh Postnatal Depression Scale (24–48 h postpartum) were 10.5 (range 0 to 25). One participant scored null, but another participant scored severely (maximum 25). Three participants scored more than 0 for the item Suicide. There were no statistically significant gender differences.

#### 3.3.2. Psychopathological Impact according to Gender at Visit 1

Table 2 indicates the psychopathological impact according to gender. Median scores in the Edinburgh Postnatal Depression Scale (1 month after the loss) were 8 (range 0 to 20). Two participants scored null but one participant still scored severely (maximum 20). None of the participants scored more than 0 for the item Suicide. There were no statistically significant gender differences.

#### 3.3.3. Grief Characteristics according to Gender

There were no significant gender differences in the total Perinatal Grief Score (PGS), nor in the characteristics of the grief experienced (Table 2). Only two participants—one father and one mother—presented with possible complicated grief.

## 4. Discussion

In summary, we found no gender differences in initial depressive EPDS symptoms (24 to 48 h post-perinatal loss) nor in depressive symptom or perinatal grief symptom scores, nor in grief levels (total scores or complicated grief) after one month following the loss.

An international survey published in 2001 [33] compared the results from 22 studies, carried out in four countries, on a wide variety of losses in pregnancy that all used the Perinatal Grief Scale as their outcome measure. Lower grief scores were consistently related to male gender, older age, shorter pregnancy, more time since the loss, mental health status, good marital relationship, good social support and a subsequent pregnancy [33]. Following these initial results, it seems that men/fathers may be relatively “protected” from the emotional impact of pregnancy loss. Our own results, with a smaller sample, did not support these relatively “protected” aspects.

Quantitative studies [28] comparing heterosexual couples’ experiences following pregnancy loss and neonatal death, suggest that men typically experience less intense and enduring levels of grief than women. However, a smaller number of studies have found similar grief intensities between men and women (as in our study), or even higher levels of grief in men [28].

Clemence Due et al. [34] detected that, despite men tending to have less intense and less enduring levels of negative psychological outcomes than women, they were more prone to engage in compensatory behaviors such as increased alcohol consumption.

Qualitative studies [34] have indicated that men often feel that their role is primarily as a ‘supporter’ to their female partner, and that this precludes recognition of their own loss. These studies also reported that men may feel overlooked and marginalized in comparison to their female partners, whose experienced pain is typically more visible [34].

Finally, Kate Louise Obst et al.’s [35] review of 46 articles (26 qualitative, 19 quantitative and one mixed-methods paper) indicated that “men’s grief experiences are highly varied, and current grief measures may not capture all of the complexities of grief for men” [35]—page 1.

Qualitative studies [35] have identified that, in comparison to women, men may face different challenges including social expectations to support female partners and a lack of social recognition for their grief and subsequent needs. It has been proposed that men may face double-disenfranchised grief in relation to the pregnancy/neonatal loss experience [35]. Unfortunately, we did not assess these aspects in our initial sample.

The same research group [28] performed an online survey on 228 Australian men aged at least 18 years whose partner had experienced a gestational loss, exploring their experiences of grief. Men experienced significant grief across all perinatal loss types, and perinatal grief was a highly individualized experience, not necessarily dependent on the gestational age. Men’s total perinatal grief scores were associated with their loss history, marital satisfaction, availability of social support, acknowledgement of their grief from family/friends, time spent bonding with the baby during pregnancy and feeling as though their role of ‘supporter’ conflicted with their ability to process their grief. Factors contributing to their grief also differed depending on their grief style [28].

Obst et al. [28,35] concluded that “there is a need to increase the accessibility of support services for men following pregnancy/neonatal loss, and to provide recognition and validation of their experiences of grief” [35]—page 1. Cohort studies are required in varied groups of bereaved men to confirm grief-predictor relationships, and to refine different models [35].

After pregnancy loss and neonatal death, men can experience high levels of grief—requiring acknowledgement and validation from healthcare professionals, family/friends, community networks and workplaces. Addressing male-specific needs, such as balancing a desire to both support and be supported, requires tailored information and support [28]. Strategies to support men should consider grief styles and draw upon father-inclusive practice recommendations [28].

In our sample, there are no gender differences in grief in these initial phases of the process. We consider that this moment is an acute traumatic stage, and we hypothesize that they are only in the first phases of grief, and that the grief’s evolution has not started yet. Our preliminary results showed that the father’s initial grief (or trauma) was variable, and that it could be as strong as that of the mother’s. Gender stereotypes of emotional communication and expression could also change over generations, and young fathers can easily show more culturally accepted emotional responses than in past generations.

The initial psychopathological impact on our sample also included some level of suicide ideation in three subjects. Several previous reviews include perinatal loss among the risk factors for suicide ideation or suicide attempts in the perinatal period [36,37,38,39]. In their scoping review of the biopsychosocial risk factors for perinatal suicidal ideation, Bright, Doody and Tuohy [36] identified traumatic experiences relating to miscarriage and pregnancy loss as an independent risk factor for suicidal ideation. These results were in the same direction as our data. Evidently, every woman (and couple) living with this loss could also have resilience and protective factors, but—in our opinion—psychological support for couples living with severe grief reactions to perinatal losses must include suicide ideation screening and management.

Spanish society provides different levels of care for the different health problems that can be associated with these losses. Perhaps—although there is little information about it—the associated physical health problems receive even more attention, or relatively even more attention than the psychological ones. On the other hand, comprehensive care for perinatal mental health problems is not yet adequately developed in all the hospitals in Spain. Despite commendable initiatives, we are still far from obtaining optimal results in psychological and psychiatric care at the perinatal level [40,41]. As the studies developed by the Spanish Charity association *Umamanita* have shown, “Many care practices that are standard in other high-income countries are not routine in Spanish hospitals. Providing such care is a relatively new phenomenon in the Spanish health system, the results provide a quality benchmark and identify a number of areas where hospitals could make improvements to care practices that should have important psychosocial benefits for women and their families“ [40]—page 1.

There is a lack of specialists and resources, but above all, there is a lack of training and—on a social level—a lack of breaking down of the associated stigma, which are the barriers that keep mothers and their partners away from seeking the care they need [3,25]. Following a systematic review of qualitative, quantitative and mixed-method studies researching parents and healthcare professionals’ experiences of care after stillbirth in high-income westernized countries [25], parents and staff together identified the need for improved training, continuity of care, supportive systems and structures and clear care pathways. Moreover, parents and staff have different points of view of other aspects of care or of their unmet needs [25]. In addition, a qualitative meta-summary of parents’ and health professionals’ experiences of stillbirth care in low- and middle-income countries (LMICs) showed that the absence of recognition worsened negative experiences of stigmatization, blame, devaluation and loss of social status. It showed that well-developed health systems are needed to address these problems, with trained staff and organizational support [42].

As Guida Rubio, President of the association of bereaved families *Anhel Vallés* says, “in our society, perinatal losses are still a “taboo”, these families are getting smaller, they are covering up their pain”, and couples are “the eternal forgotten in mourning” [43] [minutes 19:36, 26:55 and 31.03]. In fact, what different studies say is that fathers (and partners) have to face different challenges in these moments of loss—including expectations about a caring role for mothers—which further prevents them from being able to receive recognition of their needs and their right to bereavement [35,44].

In our opinion, we need to better understand the psychological evolution of couples in cases of perinatal loss without falling into preconceived ideas about the influence of gender in different samples, cultures and settings. In fact, grief reactions are influenced by the multilevel systems in which individuals are embedded [45,46,47,48,49,50]. For example, traditional Mediterranean (as Catalonian) ‘‘culturally sanctioned’’ models of grief in men are different to the mothers ‘‘culturally sanctioned’’ models of grief. However, contemporary multicultural societies (as represented in the composition of our sample) could sanction different models of grief at the same time, inside the same society. Additionally, grief processes in new young generations could also be influenced by the global social media. All of these processes could modify the traditionally ‘‘culturally sanctioned’’ Mediterranean men’s grief process.

Further research is needed in order to understand the gender differences in the impact of perinatal losses, with the aim of developing new protocols that guarantee the wellbeing of our patients.

### Limitations and Strengths

This study has several limitations, including the sample size, the recruitment characteristics and the high rate of dropouts. The initial sample included only data from 17 couples, and the one-month data only seven couples. This was because of the relatively low access to fathers’ data. We did not carry out the comparative analysis couple by couple. The main strengths of our study are the longitudinal design and the gender analysis.

## 5. Conclusions

In our sample, there were no significant gender differences in the initial impact of perinatal losses. Our data also have important limitations, but represent an unselected contemporary sample, useful to contrast with older samples. Studying the impact on both partners may be of interest in order to understand gender differences and to establish adapted therapeutic strategies. This study is ongoing and aims to address these limitations in order to detect pathways of evolution and treatment outcomes according to gender.

## Figures and Tables

**Table 1 ijerph-20-01304-t001:** Sociodemographic and clinical data of the sample (N = 46).

	Sample (N = 46)	Women (n = 29)	Men (n = 17)	*U. X^2^*	*p*
**Age (years): Median (range)**	34.0 (26–44)	33.9 (27–42)	35.0 (26–44)	271.00	0.502
**Weeks of pregnancy: Median (range)**	21.5 (13.3–40.2)	-	-	-	-
**Number of previous children: Median (range)**	1.0 (0–5)	1.0 (1–5)	1.0 (0–3)	181.5	0.836
**Marital State, n (%)**				3.13	0.077
- Single	4 (8.7)	2 (6.9)	2 (11.8)		
- Married or living in couple	41 (89.1)	26 (89.7)	15 (88.2)		
- Divorced or separated	1 (2.2)	1 (3.4)	-		
**Country of origin, n (%)**				4.939	0.764
- Spain	35 (76.1)	21 (72.4)	14 (82.4)		
- Venezuela	2 (4.3)	1 (3.4)	1 (5.9)		
- Portugal	1 (2.2)	-	1 (5.9)		
- Paraguay	3 (6.5)	2 (6.9)	1 (5.9)		
- Morocco	1 (2.2)	1 (3.4)	-		
- Equator	1 (2.2)	1 (3.4)	-		
- Estonia	1 (2.2)	1 (3.4)	-		
- Dominican Republic	1 (2.2)	1 (3.4)	-		
- Nigeria	1 (2.2)	1 (3.4)	-		
**Educational level, n (%) ***				2.786	0.426
- Without formal studies	2 (4.4)	2 (6.9)	-		
- Primary	9 (20.0)	5 (17.2)	4 (25.9)		
- Secondary	21 (46.7)	12 (41.4)	9 (56.3)		
- University	13 (28.9)	10 (34.5)	3 (18.8)		
**Work situation, n (%)**				1.795	0.773
- Unemployed, housewife or without paid work	9 (19.6)	7 (24.1)	2 (11.8)		
- Non-qualified work	3 (6.5)	2 (6.9)	1 (5.9)		
- Manual workers semi-skilled	12 (26.1)	6 (20.7)	6 (35.9)		
- Skilled manual workers or small business owners	13 (28.3)	8 (27.6)	5 (29.4)		
- Intermediate officials, owners of large businesses, university professors, etc.	9 (19.6)	6 (20.7)	3 (17.6)		
**Religious belief, n (%)**				1.284	0.733
- Cristian	16 (34.8)	10 (34.5)	6 (35.3)		
- Cristian, not practicing	13 (28.3)	7 (24.1)	6 (35.3)		
- Muslim	1 (2.2)	1 (3.4)	-		
- Not religious beliefs	16 (34.8)	11 (37.9)	5 (29.4)		
**Previous experiences of perinatal losses: Yes, n (%)**	-	9 (31.0)	-	-	-

*U*: Mann–Whitney U Test. *X*^2^: Chi Square Test. * Some data is missing.

**Table 2 ijerph-20-01304-t002:** Psychopathological characteristics of the sample (N = 46).

	Total Sample (N = 46)	Women (n = 29)	Men (n = 17)	*U. X^2^*	*p*
**Initial Edinburgh Depression Scale (24–48 h postpartum): Median (range)**	10.5 (0–25)	11.0 (0–25)	10.0 (3–24)	241.0	0.900
	**N = 18**	**n = 11**	**n = 7**		
**Edinburgh Depression Scale (one month later): Median (range)**	8.0 (0–20)	8.0 (0–20)	9.0 (2–13)	28.0	0.828
**PGS total scores (one month later): Median (range)**	35.0 (24–65)	35.0 (24–65)	32.5 (27–44)	25.0	0.425
**Active Grief * (one month after loss): n (%)**	16 (88.9)	9 (81.8)	7 (100.0)	1.226	0.359
**Complicated Grief ** (1 month after loss): n (%)**	2 (11.1)	1 (9.1)	1 (14.3)	0.003	0.641

*U*: Mann–Whitney U Test. *X*^2^: Chi Square Test. PGS: Perinatal Grief Scale. * Active Grief: When the sum of the score obtained in PGS items 1 to 4 is equal to or greater than 8 points. ** Possible Complicated Grief: When the sum of the PGS score obtained in items 5 to 16 is equal to or greater than 42 points.

## Data Availability

Not applicable.

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
