# Peer review of "Initial Impact of Perinatal Loss on Mothers and Their Partners"

_ijerph, 2023, doi:10.3390/ijerph20021304_

Round 1

Reviewer 1 Report

The article is of notable interest for the originality of its subject, since as the authors themselves refer, there is not much research on it. However, there is one point to be highlighted and that needs to be improved: Although the introduction is too brief, some aspects that are reflected in the discussion can be added in that section, and perhaps it is not adequate in that point. In fact, when talking about perinatal loss, it would be useful to indicate what the risk factors may be causing this phenomenon. However, in the discussion they comment on studies on the subject, which could go perfectly in the introduction, serving as a basis to later discuss the results found.

On the other hand, it is necessary to define the objective of the study

In relation to the sample, it is true that it is small, but it would be interesting to know the characteristics of the final sample: 29 women and 17 men. It is asumed that not all are a couple, since they don’t coincide in number. This makes it difficult to compare them by sex, but especially the relationship between couples who have suffered this loss. It is encouraged to increase the participating sample, but above all, to increase the number of couples that could facilitate the achievement of more conclusive results for the objective that is set.

Author Response

Manuscript ID

ijerph-2098411

Title

Initial Impact of Perinatal Losses on Mothers and their Partners

Dear Reviewer 1:

Thank you for all the commentaries and suggestion.

We answer all of them below. We highlighted the changes in yellow in the new version of the article.

Cordially,

The Authors

Question 1: "Comments and Suggestions for Authors: The article is of notable interest for the originality of its subject, since as the authors themselves refer, there is not much research on it. However, there is one point to be highlighted and that needs to be improved: Although the introduction is too brief, some aspects that are reflected in the discussion can be added in that section, and perhaps it is not adequate in that point. In fact, when talking about perinatal loss, it would be useful to indicate what the risk factors may be causing this phenomenon. However, in the discussion they comment on studies on the subject, which could go perfectly in the introduction, serving as a basis to later discuss the results found."

Response 1: We change the introduction and discussion in these sense.

Question 2: On the other hand, it is necessary to define the objective of the study.

Response 2: We define the objective of the study.

Question 3: "In relation to the sample, it is true that it is small, but it would be interesting to know the characteristics of the final sample: 29 women and 17 men. It is asumed that not all are a couple, since they don’t coincide in number. This makes it difficult to compare them by sex, but especially the relationship between couples who have suffered this loss. It is encouraged to increase the participating sample, but above all, to increase the number of couples that could facilitate the achievement of more conclusive results for the objective that is set."

Response 3: Yes, you are right. The initial sample included only the data from 17 couples, and the one month data only 7 couples. It was because the relative low access to father's data. We did not realize the comparative analysis couple by couple. We included that point in the limitation paragraph.

We are planned now to write two articles more. One of them analyses process satisfaction and memento mori decisions / fulfilment by gender and we will compare these data taking on account the impact over to be couple (of the same loss). It will be send in a few weeks to a specific Spanish publication (Matronas Profesión). Other future article will analyses the comparative results couple by couple, and perhaps also the qualitative information of the responses of the participants.

Thank you very much

Reviewer 2 Report

Dear authors, 

Well done! Good, first steps in research on a, as you say yourself, still very stigmatized topic. The set-up of both your study and your article is clear. Please see below some additional comments:

- in the second paragraph of your introduction, I would change the wording of ''it turns out to be a shocking life event...''; I think the loss of a child is an understandably shocking life event to all

- rephrase this sentence, this is poor English: ''Perinatal bereavement is characterized by its low visibility and recognition, although more and more protocols are being developed to address this vital moment.''

- be consistent in your use of phrase for the study participants, I would not refer to them as volunteers

-  '' We consider that this moment is an acute traumatic stage and we hypothesize that the real grief has not started yet'' > I understand what you're trying to say here, but it sounds a bit blunt. Maybe rephrase it by referring to the psychological stages of grief?

-''In our opinion, we need to better understand the psychological evolution of couples in cases of perinatal loss without falling into preconceived ideas about the influence of gender in different samples, cultures and settings.'' > I very much agree with your opinion, can you also relate it to your study? 

-You should elaborate more on your strengths and limitations and argue how it might or isn't likely to affect your study

- Final sentence of your conclusion: 'pretends to improve' should be 'will address' or 'aims to address'

I am curious to see more results from your studies, hopefully some qualitative research too! 

All the best

Author Response

Manuscript ID

ijerph-2098411

Title

Initial Impact of Perinatal Losses on Mothers and their Partners

Dear Reviewer 2:

Thank you for all the commentaries and suggestion.

We answer all of them below. We highlighted the changes in yellow in the new version of the article.

Cordially,

The Authors

Question 1: "Dear authors,  Well done! Good, first steps in research on a, as you say yourself, still very stigmatized topic. The set-up of both your study and your article is clear. Please see below some additional comments: - in the second paragraph of your introduction, I would change the wording of ''it turns out to be a shocking life event...''; I think the loss of a child is an understandably shocking life event to all".

Response 1: Modified. Thank you.

Question 2: "- rephrase this sentence, this is poor English: ''Perinatal bereavement is characterized by its low visibility and recognition, although more and more protocols are being developed to address this vital moment.''

Response 2: Modified.

Question 3: "- be consistent in your use of phrase for the study participants, I would not refer to them as volunteers.

Response 3: Modified.

Question 4: "-  '' We consider that this moment is an acute traumatic stage and we hypothesize that the real grief has not started yet'' > I understand what you're trying to say here, but it sounds a bit blunt. Maybe rephrase it by referring to the psychological stages of grief?"

Response 4: Modified.

Question 5: "-''In our opinion, we need to better understand the psychological evolution of couples in cases of perinatal loss without falling into preconceived ideas about the influence of gender in different samples, cultures and settings.'' > I very much agree with your opinion, can you also relate it to your study?.

Response 5: Yes. We introduced some proposal and references.

Question 6: -You should elaborate more on your strengths and limitations and argue how it might or isn't likely to affect your study".

Response 6: We elaborate more the section.

Question 7: "- Final sentence of your conclusion: 'pretends to improve' should be 'will address' or 'aims to address'.

Response 7: We modified it.

Question 8: "I am curious to see more results from your studies, hopefully some qualitative research too!".

Response 8: We are planned now to write two articles more. One of them analyses process satisfaction and memento mori decisions / fulfilment by gender and we will compare these data taking on account the impact over to be couple (of the same loss). It will be send in a few weeks to a specific Spanish publication (Matronas Profesión). Other future article will analyses the comparative results couple by couple, and perhaps also the qualitative information of the responses of the participants.

Thank you very much

Reviewer 3 Report

Dear authors, 

After reviewing your article entitled “Initial impact of perinatal loss on mothers and their partners”, I would like to make you some comments. I think your manuscript must be improved. I will provide you some observations I hope you can find useful. 

The introduction is too brief. You point out that this type of grief has its own characteristics, but which ones? You can explain it in more detail. Are there differences when the pregnancy has been desired? What says scientific literature to this respect? You do not explain which are the repercussions of perinatal loss in mental health. You should introduce these ideas in the text. And, I miss the objectives and hypothesis of your work. 

Your research design is not well explained. Which is the meaning of “tertiary university hospital”? Perhaps readers do not understand which are the differences with other kind of hospitals. If I am not mistaken, your design is a repeated measures design, but you do not say that. 

The sample is not described in Participants, inclusion and exclusion criteria. There is missing information about their sociodemographic characteristics. Besides, you point out that parents with too intense grief were excluded, why? I think you should explain it in more detail. 

In the Procedure section it is said that you have measured emotional state. However, you do not say which tool you use and the reasons for that. You should provide more information about it. 

Data related with the “Sociodemographic data” should be described in the Participants section. 

Instruments section must be improved. There is no information about scales version used, number of items, scores, results interpretation, reliability of the original scale and for this study. Authors should provide all this information to be possible the replication of their research. 

Information provided in Participants and Sociodemographic characteristics of the Results section should be in the previous sections given that they are not target results from this study. All this information is useful to describe the sample. In this sense, Table 1 should be moved to Participants section of the Method section. Information about differences between groups can be maintained in Results section. Moreover, I think information about all the countries, educational level, work situation, and religious belief should be provided in Table 1 to understand the results. 

Information provided in paragraphs 2 to 12 in Discussion provide theoretical information. I think this information should be moved to Introduction section. On paragraph 13 you point out that there is no differences due grief is not initiated. Why you do not collect data after the first month of the loss? Are your results similar of those reported on this paragraph? You can say something about it. On 14 paragraph you talk about emotions, but you do not provide information about it in the manuscript. I suggest to include the data related with this variable. What say the scientific literature about the information provided in this paragraph? In the following paragraph (15) you talk about limitations of the Spanish hospitals. Can you point out which ones? What are you referring to? In the same sense, I think you can provide data to support arguments presented in paragraph 16. 

In verbatim cites you should provide page numbers. 

And, at last but not least, references do not follow any style. I recommend to review it and use the journal recommended style. 

That is all from me. I hope you find my comments useful to improve your manuscript. 

Best wishes,
Dr-05

Author Response

Manuscript ID

ijerph-2098411

Title

Initial Impact of Perinatal Losses on Mothers and their Partners.

Dear Reviewer 3:

Thank you for all the commentaries and suggestions.

We answer all of them below. We highlighted the changes in yellow in the new version of the article.

Cordially,

The Authors

Question 1: " The introduction is too brief. You point out that this type of grief has its own characteristics, but which ones? You can explain it in more detail."

Response 1: Yes. We explain briefly some different characteristics related to other type of grief.

Question 2: "Are there differences when the pregnancy has been desired? What says scientific literature to this respect?"

Response 2: Excuse me, I do not find information in PubMed using the terms "desired pregnancy AND pregnancy loss", or "desired pregnancy AND perinatal loss".

Some 2013 review data showed that, in terms of risk of mental disorders, three studies showed a greater risk of mental disorders due to abortion versus miscarriage, four found no difference and two found that short-term anxiety and depression were higher in the miscarriage group (Carlo V Bellieni & Giuseppe Buonocore. Abortion and subsequent mental health: Review of the literature Psychiatry Clin Neurosci. 2013 Jul;67(5):301-10. doi: 10.1111/pcn.12067).

Obstetric risk factors for perinatal depression include unplanned or unwanted pregnancy, but I do not find data about unplanned or unwanted (undesired) pregnancy as a risk factor for perinatal loss grief. In fact, all type of perinatal losses (including voluntary abortion or abortion due to severe fetal abnormalities), appears related to emotional and psychopathological reactions, including possibly perinatal loss grief.

On the other hand, personal characteristics and specific origin of the losses could influence their emotional and psychopathological impact or the evolution of the grief.

references:

Anne Nordal Broen, Torbjørn Moum, Anne Sejersted Bødtker, Oivind Ekeberg. The course of mental health after miscarriage and induced abortion: a longitudinal, five-year follow-up study. BMC Med. 2005 Dec 12;3:18. doi: 10.1186/1741-7015-3-18.

Anne Nordal Broen, Torbjörn Moum, Anne Sejersted Bödtker, Oivind Ekeberg. Predictors of anxiety and depression following pregnancy termination: a longitudinal five-year follow-up study. Acta Obstet Gynecol Scand. 2006;85(3):317-23. doi: 10.1080/00016340500438116.

Alba Calderer, Noemí Obregón, Jesús Vicente Cobo, Josefina Goberna4. Muerte perinatal: acompañamiento a mujeres y parejas [Perinatal death: accompanying women and couples][In Spanish]. Matronas Prof. 2018; 19(3): e41-e47

Kirsty Ryninks, Megan Wilkinson-Tough, Sarah Stacey, Antje Horsch. Comparing posttraumatic growth in mothers after stillbirth or early miscarriage. PLoS One. 2022 Aug 8;17(8):e0271314. doi: 10.1371/journal.pone.0271314. eCollection 2022.

Annsofie Adolfsson and Per-Göran Larsson. Applicability of general grief theory to Swedish women's experience after early miscarriage, with factor analysis of Bonanno's taxonomy, using the Perinatal Grief Scale. Ups J Med Sci. 2010 Aug; 115(3): 201–209. doi: 10.3109/03009731003739851

Marguerite Maguire, Alexis Light, Miriam Kuppermann, Vanessa K Dalton, Jody E Steinauer, Jennifer L Kerns. Grief after second-trimester termination for fetal anomaly: a qualitative study. Contraception. 2015 Mar;91(3):234-9. doi: 10.1016/j.contraception.2014.11.015. Epub 2014 Dec 12.

Oya Güçlü, Güliz Åženormanci, Abdullah Tüten, Koray Gök, Ömer Åženormanci. Perinatal Grief and Related Factors After Termination of Pregnancy for Fetal Anomaly: One-Year Follow-up Study. Noro Psikiyatr Ars. 2021 May 17;58(3):221-227. doi: 10.29399/npa.25110. eCollection 2021.

We include some expression and references in the text.

Question 3: "You do not explain which are the repercussions of perinatal loss in mental health. You should introduce these ideas in the text."

Response 4: I resume the repercussions: "The psychological consequences include a significant presence of anxiety symptoms and depression, and can even lead to post-traumatic stress disorder and even more severe mental disorders".

Question 5: "And, I miss the objectives and hypothesis of your work.".

Response 5: We included them.

Question 6: "Your research design is not well explained. Which is the meaning of “tertiary university hospital”? Perhaps readers do not understand which are the differences with other kind of hospitals."

Response 6: Tertiary university hospital refers (in Spanish context) to a Hospital linked to a University but at the high level of assistance, without no other hospital to assist in difficult cases. I erase the word "tertiary".

Question 7:" If I am not mistaken, your design is a repeated measures design, but you do not say that.".

Response 7: Yes. We modified it.

Question 8: "The sample is not described in Participants, inclusion and exclusion criteria."

Response 8: We included the inclusion an exclusion criteria.

Question 9: "There is missing information about their sociodemographic characteristics."

Response 9: We modified Table 1 and we included more specific sociodemographic characteristic.

Question 10: "Besides, you point out that parents with too intense grief were excluded, why? I think you should explain it in more detail."

Response 10: Yes. Because ethical reasons. In one couple case, they did not want to communicate with the medical staff at these moment. We included these information in the article.

Question 11: "In the Procedure section it is said that you have measured emotional state. However, you do not say which tool you use and the reasons for that. You should provide more information about it.

Response 11: We assess the emotional state clinically, we did not use a specific scale at the moment.

Question 12: "Data related with the “Sociodemographic data” should be described in the Participants section."

Response 12: We include a reference to Table 1.

Question 13: "Instruments section must be improved. There is no information about scales version used, number of items, scores, results interpretation, reliability of the original scale and for this study. Authors should provide all this information to be possible the replication of their research."

Response 13: We included information.

Question 14: "Information provided in Participants and Sociodemographic characteristics of the Results section should be in the previous sections given that they are not target results from this study. All this information is useful to describe the sample. In this sense, Table 1 should be moved to Participants section of the Method section. Information about differences between groups can be maintained in Results section."

Response 14: We move the Table 1 to the Participants section.

Question 15: "Moreover, I think information about all the countries, educational level, work situation, and religious belief should be provided in Table 1 to understand the results."

Response 15: We included all of them.

Question 16: "Information provided in paragraphs 2 to 12 in Discussion provide theoretical information. I think this information should be moved to Introduction section.

Response 16: We move it.

Question 17: "On paragraph 13 you point out that there is no differences due grief is not initiated. Why you do not collect data after the first month of the loss? Are your results similar of those reported on this paragraph? You can say something about it.

Response 17: The study is prospective and it is not ended. We have some information about evolution longer than one month only in a few couples. We decide to show now the first data and we intend to recruit more couples for a future publication.

Question 18: "On 14 paragraph you talk about emotions, but you do not provide information about it in the manuscript. I suggest to include the data related with this variable. What say the scientific literature about the information provided in this paragraph?"

Response 18: The researchers attach the participants during the answers in order to respond to doubts and to give support and relief in case of emotional reactions. The scale we used for the psychopathological evaluation was the Edinburg EPDS. The initial emotional state was evaluated clinically for the researchers based on the communication and behaviors showed. After one month of the loss, we uses the Perinatal Grief Scale to evaluate the emotional state. We do not use other scales of emotion. We included some more data about the cultural aspects of grief in the following paragraphs.

Question 19: "In the following paragraph (15) you talk about limitations of the Spanish hospitals. Can you point out which ones? What are you referring to?"

Response 19: We have some relevant data about the limitations of the management of pregnancy loss in Spain. Studies developed by the Spanish Charity association Umamanita showed, “Many care practices that are standard in other high-income countries are not routine in Spanish hospitals. Providing such care is a relatively new phenomenon in the Spanish health system, the results provide a quality benchmark and identify a number of areas where hospitals could make improvements to care practices that should have important psychosocial benefits for women and their families“.

References, among others:

- Cassidy, P.R. Care quality following intrauterine death in Spanish hospitals: results from an online survey. BMC Pregnancy and Childbirth volume 18, Article number: 22 (2018). The verbatim cite is from the abstract (Page 1).

- Cassidy, P.R. The Disenfranchisement of Perinatal Grief: How Silence, Silencing and Self-Censorship Complicate Bereavement (a Mixed Methods Study). OMEGA - Journal of Death and Dying, 0(0). https://doi.org/10.1177/00302228211050500

Question 20: "In the same sense, I think you can provide data to support arguments presented in paragraph 16."

Response 20: We include data from meta-analysis and bibliography.

Question 21: "In verbatim cites you should provide page numbers".

Response 21: We do it for the Cassidy (2018) reference (page 1), for Obst et al. (2020) (page 1) and for the minutes of the video from Guida Rubio (minutes 19:36, 26:55 and 31.03).

Question 22: "And, at last but not least, references do not follow any style. I recommend to review it and use the journal recommended style."

Response 22: Yes. We do it, including all the old references and the new references proposed or suggested by all the Reviewers (changes marked in red).

We revised the English expressions in both the abstract and text.

Yes, we find Your comments useful to improve our manuscript.

Thank you very much

Round 2

Reviewer 3 Report

Dear authors, 

After reviewing for the second time your article entitled “Initial impact of perinatal loss on mothers and their partners”, I think there are still some aspects that could be improved. Anyway, I think is an interesting topic and, despite you do not find significative differences, it deserves to be known by the scientific community. 

In this case, all my comments are highlighted in the PDF file of your manuscript. 

That is all from me. I hope you find my comments useful to improve your manuscript. 

Best wishes,
Dr-05

Author Response

Dear Reviewer:

I send you the article with the response to Your comments.

The new version with all Your comments is ready.

Thank you,

Jesus Cobo
